# Evaluation of Ferromagnetic Steel Hardness Based on an Analysis of the Barkhausen Noise Number of Events

**DOI:** 10.3390/ma13092059

**Published:** 2020-04-29

**Authors:** Maciej Roskosz, Krzysztof Fryczowski, Krzysztof Schabowicz

**Affiliations:** 1Faculty of Mechanical Engineering and Robotics, AGH University of Science and Technology, aleja Mickiewicza 30, 30-059 Kraków, Poland; 2Faculty of Energy and Environmental Engineering, Silesian University of Technology, ul. Akademicka 2A, 44-100 Gliwice, Poland; krzysztof.fryczowski@polsl.pl; 3Faculty of Civil Engineering, Wrocław University of Technology, Wybrzeże Wyspiańskiego 27, 50-370 Wrocław, Poland; k.schabowicz@pwr.edu.pl

**Keywords:** hardness, Barkhausen noise, number of events, non-destructive testing inverse problem

## Abstract

Measurements are carried out of the Barkhausen noise (BN) and hardness on specimens where changes in hardness were obtained due to strain hardening (S235 and DC01 steels) and due to thermochemical treatment (AMS 6414 steel). A method is presented of processing the recorded BN signal to extract diagnostic information. The BN number of events is selected as the signal characteristic property to develop relevant correlations. A new methodology is presented for the development of correlations between the Barkhausen noise number of events and hardness. A possibility is indicated of developing correlations with a high R2 determination coefficient. The method limitations are specified.

## 1. Introduction

Hardness is a property that cannot be defined unequivocally. In fact, many other properties are measured in hardness tests. The tests can be classified depending on the material resistance observed during the testing. Static methods of hardness measurement consist of pushing the indenter into the tested material beyond the yield point to cause plastic strains. In such methods, hardness can be defined as the measure of the material strength to plastic strains arising due to the indenter pressing force.

During a traditional hardness measurement, the hardness testing machine indenter causes plastic strain to the surface domain where the measurement is performed. For some elements, such damage to the surface is unacceptable. For others, plastic deformation of the surface is not an operating problem but a diagnostic one: it is impossible to repeat the hardness test in exactly the same location under the same conditions. These obstacles can be overcome using so-called inverse problems of non-destructive testing, where mechanical properties are determined based on other physical quantities.

In the testing presented herein, this physical quantity is the Barkhausen noise (BN), which is created due to a time-variable impact of a magnetic field on a ferromagnetic. The variable magnetic field causes the stochastic displacement of domain walls, which involves a sudden change in magnetization detectable by the measuring sensor. More details about the theory of the Barkhausen noise can be found in several models [1,2,3].

The Barkhausen noise is used as a diagnostic signal to determine the material stress-and-strain state [4,5,6,7], the state of the microstructure [8], defects in surface treatment [9], and hardness [5,9,10,11,12,13,14,15].

Ortega-Labra et al. [4] presented a novel system for measuring stress applied to a pipe section using the rotational continuous magnetic Barkhausen noise (MBN). The presented results show that the shape and the amplitude of the MBN angular dependence change with an increase in applied stress. The outcome reveals that the applied stress value could be estimated using the amplitude of the angular dependence of the measured MBN signal. Additionally, the shape of the angular dependence is related to the applied stress direction. This suggests that using a proper procedure, the direction of the applied stress could also be estimated based on the MBN.

Liu et al. [5] presented the application of a multifunctional magnetic testing technique in the quantitative prediction of tensile stress, surface hardness, and the case depth in medium-carbon steel rods. Simultaneous measurements were carried out on the MBN signal, the tangential magnetic field, and the magnetic hysteresis curve in tensioned steel rods with different depths of the surface-hardened layer. A total of 16 feature parameters were extracted from four types of signal patterns as candidate inputs of the back-propagation neural network. Very promising results were obtained, where the prediction model error was less than 5.3% for tensile stress, 5% for the case depth, and 0.22% for surface hardness.

In [6], skewness is utilized as a feature for applied stress detection and compared with the root-mean-square (RMS) peak value and peak position, which are the most common features of the MBN signal. Different levels of compressive and tensile stresses are applied to En36 steel specimens to evaluate the new feature effectiveness. Its non-linear behavior and sensitivity are compared with the RMS peak value and peak position. The domain wall energy and the eddy current damping are discussed as the cause of the phenomenon.

In [9], He et al. presented a review of non-destructive testing methods for the metallic grinding burn detection. One of them makes use of measurements of the magnetic Barkhausen noise (MBN) and the magnetoelastic acoustic emission (MAE). It is found that the BN method itself produces only relative results that need to be compared with calibration blocks to define certain limits for rejection. Several aspects for further investigation are defined, such as the relationship between the BN signal multiparameters and the measured material properties (stress state, hardness), the impact factors of BN measurements including microstructural domain walls, and macroscopic operations, such as the influence of the holding force, the contact angle between the probe and the workpiece, and the probe moving speed.

Wilson et al. [10] studied P9 and T22 steel specimens subjected to different kinds of heat treatment used to modify the microstructure. Magnetic hysteresis loops and the Barkhausen noise were measured and compared to the Vickers hardness number. It is demonstrated that a rise in hardness involves a rise in coercivity, a change in the location of the maximum value of the Barkhausen noise envelope, and a drop in magnetic permeability.

Moorthy et al. [11] examined the impact of a post-hardening change in the specimen structure on the Barkhausen noise envelope on the carburized layer of steel EN36. Several frequencies of the excitation field were applied, and a frequency analysis of the measured signal was conducted in different ranges. It is proved that the envelope maximum value correlates with the hardness measured in the layer to the depth of 100 μm.

The aim of the studies presented in [12] was to use the Barkhausen noise to characterize hardness in steel AISI 4140 subjected to the Jominy test. Various types of time-frequency representations were used: the spectrogram, the Wigner–Ville distribution, the Capongram, the ARgram, the wavelet transform scalogram, and the Mellingram. It is proved that due to the Barkhausen noise non-stationary character, time-frequency representations are a rich source of information. A method is proposed of extracting information from these representations by creating an algorithm reducing the number of variables from 3 (time, frequency, amplitude) to 2 (time or frequency and amplitude).

In [13], O’Sullivan et al. measured magnetoacoustic emission (MAE) and the Barkhausen noise (BN) to determine the properties of AISI 430 steel specimens subjected to heat treatment to change their microstructure. Parameters such as the RMS value of the Barkhausen noise and the magnetoacoustic emission absolute value proposed by the authors were compared e.g., to the Vickers hardness number. It is shown that a rise in plastic strain causes a drop in the BN and MAE and that the parameters are linearly dependent on the Vickers hardness number.

Franco et al. [14] present relations between the Barkhausen noise parameters and Rockwell hardness using SAE 4140 and SAE 6150 steel specimens subjected to the Jominy hardenability test. The Barkhausen noise is described using the maximum value, the location, and the RMS value of the envelope. A linear relation is found between all parameters of the Barkhausen noise and Rockwell hardness.

In [15], many properties of the Barkhausen noise are analyzed, and a method is proposed of their selection and mutual complementation for the purposes of hardness evaluation. An algorithm is presented for the analysis of the signal main properties. Combined with the method of the analysis of the correlations between them, the algorithm is used to select the optimal properties. Multidimensional linear regression with selected properties is applied, and a statistical linear model of merging BN properties is built to forecast the material hardness.

This paper presents the results of testing aiming to develop diagnostic correlations between hardness and the Barkhausen noise total number of events *NoE_TOT_* BN. A good correlation between the two quantities would enable periodic and multiple measurements of hardness in the same location, which cannot be done using traditional hardness measurement methods. Two cases of differences in the material hardness are considered. One is a change in hardness due to cold plastic strain and the other is the change caused by thermochemical treatment.

*NoE_TOT_* is selected as the diagnostic signal property to be analyzed because the authors already obtained promising results from their previous preliminary research on hardness evaluation [16,17] and from their comprehensive studies on the assessment of the state of active stresses in steel elements [7]. The following quantities are analyzed in [7,16,17]: the BN RMS value of voltage (*U_RMS_*) [18,19,20], the BN energy (*E_BN_*) [20,21], characteristic parameters related to the BN signal envelope [22,23], and the distribution of the BN total number of events (*NoE_TOT_* BN) [24]. The best determination coefficients *R*^2^ and the smallest standard deviations are obtained for the *NoE_TOT_* BN. Another issue, not taken up in this paper, is the analysis of the intensity of changes in the features describing the Barkhausen noise that occur due to changes in hardness or in the stress-and-strain state. The signal feature that changes the most intensely is the one best suited to enable the development of a diagnostic correlation. The issue is related to digital processing of the signal on the one hand, and to analysis of the variability of the signal individual features on the other. Such analyses are presented for example in [25], where a comparison is made between changes in the Barkhausen noise profile skewness and the noise signal RMS value caused by active compressive and tensile stresses. Within certain stress ranges, the sensitivity of skewness is higher than that of the RMS value, which makes the former a better diagnostic signal.

Among others, the Barkhausen noise depends on the state of the material microstructure and on the material stress state. The impacts can be separated from each other assuming that the two factors are not changed at the same time. Changes in hardness due to strain hardening are accompanied by changes in internal (residual) stresses. Similarly, changes in hardness due to thermochemical treatment involve, apart from changes in the specimen microstructure, changes in the level of residual stresses. In the presented testing, the impact of internal stresses was not taken into consideration in the analysis of results. In consequence, the developed diagnostic correlations are less universal because they are related to the structural properties of specimens.

## 2. Measuring Apparatus

The Barkhausen noise was measured using the MEB4-C system made by the Mag-Lab s.c. company (Gdańsk, Poland). This is an independent system of measuring, analyzing, and recording the Barkhausen noise (BN) voltage signals. The system diagram is shown in Figure 1. The measurements were performed using a surface measuring head with systems of the electromagnet, Barkhausen noise detection, and magnetization control. The head diagram is presented in Figure 2. The system enables configuring settings in the range of the magnetizing current amplitudes, the rate of changes in the magnetizing current (the magnetizing current frequency), and the measured signal amplification.

## 3. BN Signal Analysis

The basic unit of information is a single cycle of changes in magnetization. The cycle is presented in detail in Figure 3. Two halves of such a full cycle can be distinguished: one in which the magnetization current intensity diminishes (marked as *I*↘) and the other in which the current intensity rises (marked as *I*↗).

To obtain information from a single cycle of changes in magnetization, the signal is processed in different ways. The processing method produces values of individual properties of the Barkhausen noise. For a multiple number of cycles of changes in magnetization, a multiple number is obtained of values of the quantity for which a statistical analysis has to be carried out. The analysis gives the average value of the quantity and the value of standard deviation being a variability measure of the BN signal analyzed properties.

The measured signal of the Barkhausen noise is a set of voltage pulses, among which so-called events can be distinguished. It is assumed that for a set value of threshold voltage *U_g_*, the event occurrence is determined based on three subsequent signal samples with a value exceeding the threshold voltage, where the value of the second signal is higher than the value of the first and third (cf. Figure 4). Detected events are summed up, which gives the distribution of the total number of events—*NoE_TOT_*—depending on threshold voltage *U_g_* (cf. the example presented in Figure 5). The threshold voltage-dependent *NoE_TOT_* distribution is obtained for each cycle of changes in magnetization. In the case of a multiple number of cycles of changes in magnetization, averaged *NoE_TOT_* distributions are determined together with statistical features, such as standard deviation.

The relation between the number of events (*NoE_TOT_*) and the corresponding values of hardness were analyzed for individual values of discrimination voltage $U_{g_{j}}$ using a linear regression model. The analysis resulted in the value of determination coefficient *R*^2^ for each value of discrimination voltage $U_{g_{j}}$. The value reflects the quality of the goodness of fit of the linear model developed based on input data. A value of determination coefficient *R*^2^ equal or higher than 0.8 proves a good fit of the linear regression model.

Additionally, the resolution coefficient (*RE_COF._*) was defined. The coefficient makes it possible to find the discrimination voltage value for which the best resolution is obtained, where the resolution is defined for a given value of discrimination voltage $U_{g_{j}}$ as the quotient of the difference between the maximum and the minimum value of the obtained total number of events ($NoE_{TOT}{}_{max}\left(U_{g_{j}}\right)$ and $NoE_{TOT}{}_{min}\left(U_{g_{j}}\right)$, respectively) and the sum of the standard deviation values $SD_{i}\left(U_{g_{j}}\right)$ corresponding to individual points of the hardness measurement.
(1)$$RE\left(U_{g_{j}}\right)=\frac{NoE_{TOT}{}_{max}\left(U_{g_{j}}\right)-NoE_{TOT}{}_{min}\left(U_{g_{j}}\right)}{\sum_{i}^{n}SD_{i}\left(U_{g_{j}}\right)}$$

To normalize the resolution coefficient (*RE*) results with the determination coefficient (*R*^2^) varying in interval <0,1>, the normalized coefficient of resolution (*RE_COF._*) was introduced. The coefficient is defined as:(2)$$RE_{COF.}\left(U_{g_{j}}\right)=\frac{RE\left(U_{g_{j}}\right)}{max\left(RE(U_{g_{j}}\right)}.$$

The best correlation occurs for the discrimination voltage for which maximum values of *R*^2^ and *RE_COF_* are obtained.

## 4. Testing of Specimens with Plastic Strain

Cold plastic strain increases dislocation density (by 4 to 5 orders of magnitude) [26], which involves the material hardening due to the decrease in distances between dislocations and the increasing forces of their interaction. Due to the smaller distance, the dislocations block each other, and their further displacement (further deformation of the material) is only possible if a higher external stress is applied; the phenomenon is referred to as strain hardening. A plastically deformed material is characterized by increased strength properties (hardness, yield point) with simultaneously decreased plastic properties (elongation, narrowing) [26].

### 4.1. Testing Details

The tests were performed on plate specimens made of S235 steel (according to Standard EN 10025) and DC01 steel (according to Standard EN 10130). The steel grades properties are listed in Table 1.

The specimens were loaded on a Galdabini Sun 10P tensile testing machine (Galdabini, Cardano al Campo (VA), Italy). They were repeatedly subjected to uniaxial tension in such a manner that with every loading cycle, there was a slight increment in plastic strain. When the ultimate strength value was exceeded, a narrowing started to form in the specimen. The loading process was discontinued the moment that the narrowing was visible enough to detect through a visual inspection. Due to the material hardening, a change in hardness occurs in the plastically deformed area. Non-uniform plastic strain leads to the creation of a specific hardness profile on the specimen surface.

For the measuring apparatus described in Section 2, the details of the configurations used during the BN measurements are presented in Table 2.

At the preliminary stage of the testing, measurements were carried out changing the magnetization direction every 30°; angles 0° and 180° correspond to the load direction. Example results are presented in Figure 6. They confirm the literature data [27,28] that prove that the signal maximum and minimum values are obtained for the maximum strain direction and for the normal direction, respectively.

In the engineering practice, diagnostic correlations should be independent of the directions of principal residual stresses in the material, which are usually unknown. One proposal for a simplified solution is the absolute value of the Barkhausen noise properties:(3)$$V=\sqrt{V_{X}^{2}+V_{Y}^{2}}$$
where *V_X_* and *V_Y_*, respectively, are the property values for two directions of the magnetizing field application (e.g., *E_BN_* or *NoE_TOT_*) for which the most extreme values are obtained.

Considering the above, the Barkhausen noise was measured only in two directions perpendicular to each other, i.e., in direction *X*, parallel to the load direction, and in direction *Y*, normal to the load direction.

After the Barkhausen noise was measured in the same measuring points, HV5 hardness measurements were performed using the Krautkramer TIV hardness tester (General Electric, Boston, MA, USA). The measurements were carried out along a measurement line being the specimen axis of symmetry. Example testing results for plastically deformed specimens made of S235 and DC01 steel are shown in Figure 7 and Figure 8, respectively.

### 4.2. Analysis of Results

According to the measurement methodology presented in Section 3, the analysis covers the results obtained in two directions perpendicular to each other and their absolute value. The distributions of determination coefficient *R*^2^ of the linear correlation between *HV5* hardness and the total number of events *NoE_TOT_* and of resolution coefficient *RE_COF_* as a function of threshold voltage *U_g_* for S235 steel are presented in Figure 9a–c for the parallel direction, the normal direction, and the absolute value, respectively. In all three charts (Figure 9a–c), the curves illustrating the history of determination coefficient *R*^2^, after the initial phase of a clear increase (in the range of *U_g_* from 0 to about 0.4 V), look quite similar; for higher voltage values, there are no essential changes. The distributions of coefficient *RE_COF_* are characterized by a global maximum falling on *U_g_* values ranging from 0.5 to 0.6 V. These maxima decide the selection of the discrimination voltage for which the linear correlations between *NoE_TOT_* and HV5 are developed and presented in Figure 10a for the parallel direction, in Figure 10b for the normal direction, and in Figure 10c for the absolute value. The values of determination coefficient *R*^2^ differ considerably for the two directions and total about 0.7 for the parallel direction (which is too low for practical use) and about 0.92 for the normal direction. However, in the analyzed case, the impact of the parallel direction of magnetization on the value of the linear correlation determination coefficient *R*^2^ for the absolute value (*R*^2^ = 0.914) is very slight, and the correlation can be used in practice.

The distributions of determination coefficient *R*^2^ of the linear correlation between *HV5* hardness and the total number of events *NoE_TOT_* and of resolution coefficient *RE_COF_* as a function of threshold voltage *U_g_* for DC01 steel are presented in Figure 11a–c for the parallel direction, the normal direction, and the absolute value, respectively. In all three charts below, the curves illustrating the history of determination coefficient *R*^2^, after the initial phase of a clear increase (in the range of *U_g_* from 0 to about 0.5 V), look quite similar; for higher voltage values, there are no essential changes. A characteristic feature of the *RE_COF._* distributions is that the first local maximum falls within the range of voltage *U_g_* variability where the determination coefficient *R*^2^ takes low values. It is only the values of the second local maximum that decide the selection of the discrimination voltage for which the linear correlations between *NoE_TOT_* and *HV5* are developed and presented in Figure 12a for the parallel direction, in Figure 12b for the normal direction, and in Figure 12c for the absolute value.

The values of determination coefficient *R*^2^ for the two directions of magnetization vary from about 0.73 to about 0.8, which is too low to use in practice. Finding the absolute value improves the situation only slightly, raising coefficient *R*^2^ for optimal discrimination voltage *U_g_* to about 0.8, which is still just on the limit of possible practical use.

In the presented results of the testing, higher *NoE_TOT_* values correspond to higher hardness (higher degree of plastic strain), which is also observed for the values of other properties of the Barkhausen noise (Figure 7 and Figure 8) [7,16,17]. However, this is not a permanent trend that occurs for every material. S235JRG2 steel is investigated in [29], and it is found that a rise in the plastic strain degree involves a decrease in the BN *U_RMS_* value measured in the load direction and an increase in the value determined in the normal direction. It is also found that the main reason for the observed changes is the number of dislocations causing non-homogeneous microdeformations. The stress field around dislocations interacts with all suitably oriented displacements of domain walls. If the number of displacements is reduced, the values of the BN properties are reduced, too. The authors of [29] state that this makes it possible to account for the drop in the BN property value measured in the load direction, but it still cannot explain the rise of the value in the normal direction. A similar trend of changes is observed in [13]. In [30], Fe–2%Si alloy is investigated using an enveloping coil, which allows the assumption that the magnetization direction is the same as the direction of the load. It is found that a rise in plastic strain involves a rise in the integrals of *U_b_* voltage. This trend is opposite to the one observed in [29]. The results of the testing of plastically deformed Armco iron presented in [31] show a non-uniform trend of changes in *U_b_* voltage integrals. After an initial increase with a rise in plastic strain, if the strain value exceeds about 10%, a further rise in strain involves a decrease in the values. Another essential factor is additionally found in [32]: the impact of the specimen loading history (the method of achieving plastic strain). In the case of specimens for which plastic strain was achieved in a single loading cycle, the BN properties decreased with a rise in plastic strain. For specimens where subsequent plastic strain values were achieved gradually to the value of about 1%, the values of the BN properties increased. A further rise in strain caused a gradual drop in the values of the BN properties. However, the drop never reached the level measured for specimens where strain was achieved in a single loading cycle.

The analysis of the phenomena of interaction between a change in the degree of plastic strain and dislocation density is a complex process that requires many more microscopic and macroscopic studies on the changes that arise in the domain structure of polycrystalline ferromagnetic materials due to a complex state of stresses and strains.

## 5. Testing of Specimens Subjected to Thermochemical Treatment

### 5.1. Testing Details

The testing was carried out on a set of three specimens made of AMS 6414 steel subjected to carburization in an atmosphere with a different concentration of carbon and then to hardening [33]. The process resulted in specimens with a different degree of hardness and a constant thickness of the carburized layer of 1.2 mm. The surface of the specimens was prepared for the testing by grinding. The core of the specimens has a martensitic structure. As for the layers, an increase in martensite refinement can be observed with a rise in hardness. This is because the number of locations of martensite nucleation during the hardening process increases with a rise in the content of carbon in the carburized layer. As a result, the growth of martensite is reduced. Moreover, cementite precipitates can be found in the hardest layer [33].

The Barkhausen noise initial measurements were performed so that an angular distribution of properties could be obtained, as presented in Figure 6. The results were analyzed, and the directions of magnetization parallel with and normal to the direction of grinding traces were adopted as the measurement directions (Figure 13). Measurements were carried out for three configurations of the measuring device settings, the details of which are listed in Table 3.

Microhardness on the surface of carburized specimens was measured using a Zwick Roell microhardness tester. The measurement results are listed in Table 4.

### 5.2. Analysis of Results

Averaged *NoE_TOT_* distributions were obtained using the data-analysis procedure described in Section 3 for three measuring configurations and two magnetization directions perpendicular to each other. The absolute value was also determined according to Relation (2). For Configuration C1, distributions of *R*^2^ as a function of *U_g_*, after strong oscillations in the *U_g_* range from 0 to about 0.4 V, stabilize, taking values close to 1 for the parallel direction (Figure 14a) and for the absolute value (Figure 14c), and values oscillating between 0.9 and 1 for the normal direction (Figure 14b). The selection of discrimination voltage *U_g_*, for which the correlations presented in Figure 15a–c were developed, is decided by the global maximum of the *RE_COF_* distribution. In the case under analysis, the maximum of voltage *U_g_* is about 0.8 V. The obtained values of determination coefficient *R*^2^ take values higher than 0.97, which proves a very good correlation.

For Configuration C2, the distributions of *R*^2^ depending on *U_g_* (Figure 16a–c) are characterized by considerable variability with two local maxima reaching the value of 1. The *RE_COF_* distributions also demonstrate high variability and have a global maximum that decides the value of discrimination voltage *U_g_* for which the correlations shown in Figure 17a–c were developed. For Configuration C2, the voltage values are higher compared to Configuration C1 (from 1.12 to 1.22 V). The obtained values of determination coefficient *R*^2^ take lower values compared to Configuration C1. They are included in the range from about 0.89 to about 0.96, which proves a good correlation.

For Configuration C3, the determination coefficient in the *U_g_*-dependent distributions of *R*^2^ (Figure 18a–c) takes values higher than 0.95 for the *U_g_* range from 0.1 to 0.4 V. This is also the range for which the *NoE_TOT_* values are higher than 0.

The *RE_COF_* distributions for this voltage range have a local maximum that decides the selection of discrimination voltage *U_g_* (from 0.1 to 0.16 V) for which the correlations presented in Figure 19a–c were developed. The values of determination coefficient *R*^2^ are included in the range from about 0.97 to about 0.987, which proves a very good correlation.

In each of the configurations under analysis, a rise in hardness involves a drop in the number of events (*NoE_TOT_*). For the tested specimens, martensite refinement increases with a rise in hardness, which means that a smaller number of events (*NoE_TOT_*) corresponds to a structure with a smaller grain size. Similar results are obtained in [34,35], whereas completely opposite relations are found in [8]. This proves that the Barkhausen noise affects not only the grain size but also the nature of grain boundaries and their mutual orientation [35]. Moreover, for a specific configuration of factors affecting the Barkhausen noise, a repeatability of results is achieved. The electromagnetic phenomena occurring in the microscale due to the effect of the size of the measuring converters make use of diagnostic relations developed based on macroscale parameters (averaged information on the material state from the macro area). Compared to the conclusions presented in [36], the diagnostic information obtained in this manner is both repeatable and reliable.

## 6. Summary and Conclusions

The Barkhausen noise and *HV* hardness were tested in specimens subjected to plastic strain and thermochemical treatment. The measurement data were processed numerically, which resulted in distributions of the total number of events (*NoE_TOT_*) as a function of discrimination voltage *U_g_*. Next, the linear correlation determination coefficient (*R*^2^) and the resolution coefficient (*RE_COF._*) introduced by the authors were analyzed in parallel to find the optimal correlation. The obtained results prove the great potential of using the Barkhausen noise total number of events (*NoE_TOT_* BN) for the development of correlations enabling hardness determination. Distributions of the total number of events (*NoE_TOT_*) in selected intervals of threshold voltage *U_g_* make it possible to develop correlations characterized by a high determination coefficient *R*^2^ in the case of changes in hardness due to both strain hardening and thermochemical treatment.

A new methodology was developed for the determination of diagnostic correlations to solve inverse problems of non-destructive testing consisting of hardness evaluation based on the Barkhausen noise total number of events (*NoE_TOT_* BN). However, it has to be emphasized that if these correlations are to be applied, factors that affect the diagnostic signal, such as the surface shape and roughness or the element edge impact, must be taken into account. The factors are described in detail in [7,17]. The sampling frequency can also play a significant role, because some pulses could overlap in the time scale.

Compared to classic static and dynamic methods of hardness evaluation using an indenter, the developed method of measuring hardness based on the total number of events (*NoE_TOT_*) of the Barkhausen noise causes no plastic strain on the tested surface and makes it possible to perform hardness measurements in the same place repeatedly.

Additionally, it is possible to utilize other properties of the Barkhausen noise to develop multi-property correlations [15], where each property will have a specific weight established based on an analysis of the determination coefficient (*R*^2^) and the resolution coefficient introduced by the authors (*RE_COF._*).

The authors, as well as other scientists [37], are aware of the deficit in basic research on the relations between changes in a broadly understood state of the material and changes in electromagnetic properties, both at the micro and macroscopic scale. Another important research issue that emerged while analyzing the testing results is the impact of the plastic strain process (whether it proceeds continuously or in stages) on changes in parameters of the Barkhausen noise quantitative description. The results of such research would enable more confident actions in inverse problems of non-destructive testing using electromagnetic phenomena.

## Figures and Tables

**Figure 1 materials-13-02059-f001:**
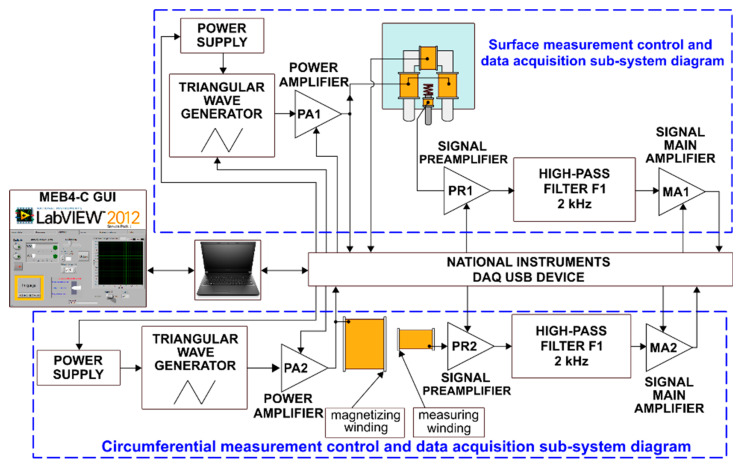
Schematic diagram of the MEB4-C system.

**Figure 2 materials-13-02059-f002:**
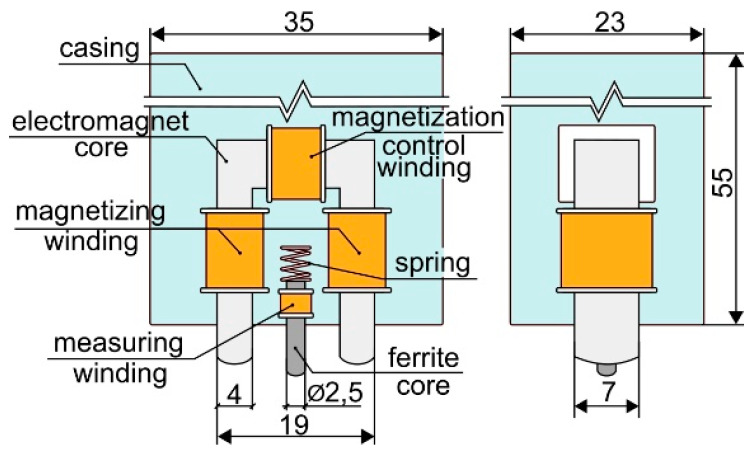
Surface measuring probe.

**Figure 3 materials-13-02059-f003:**
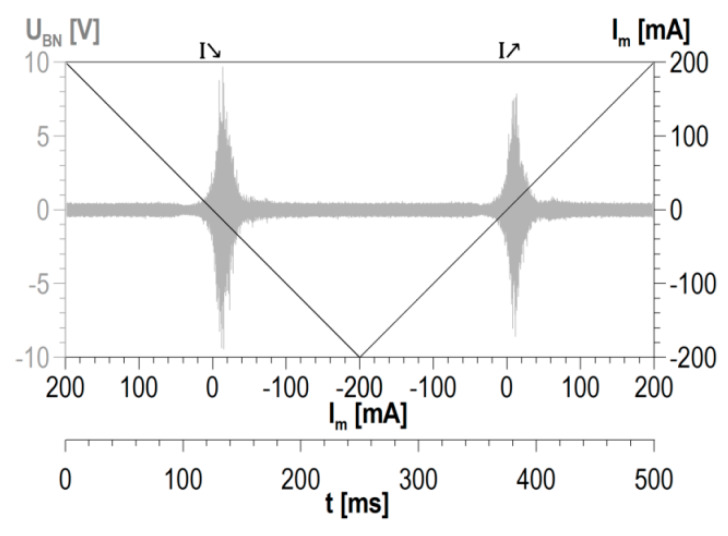
Detailed description of a single cycle of changes in magnetization.

**Figure 4 materials-13-02059-f004:**
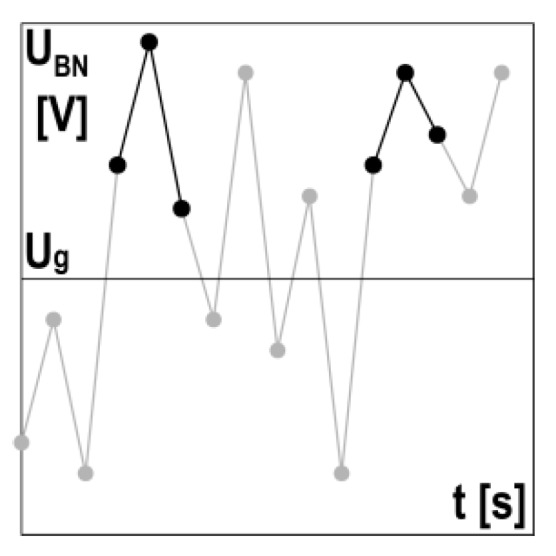
Method of event determination above the set value of threshold voltage *U_g._*

**Figure 5 materials-13-02059-f005:**
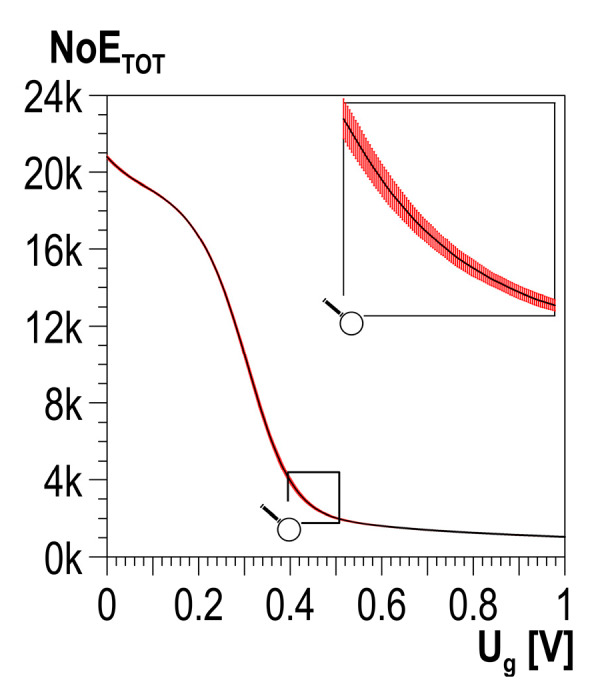
Averaged distribution of the total number of events (*NoE_TOT_*) with standard deviation values (marked in red).

**Figure 6 materials-13-02059-f006:**
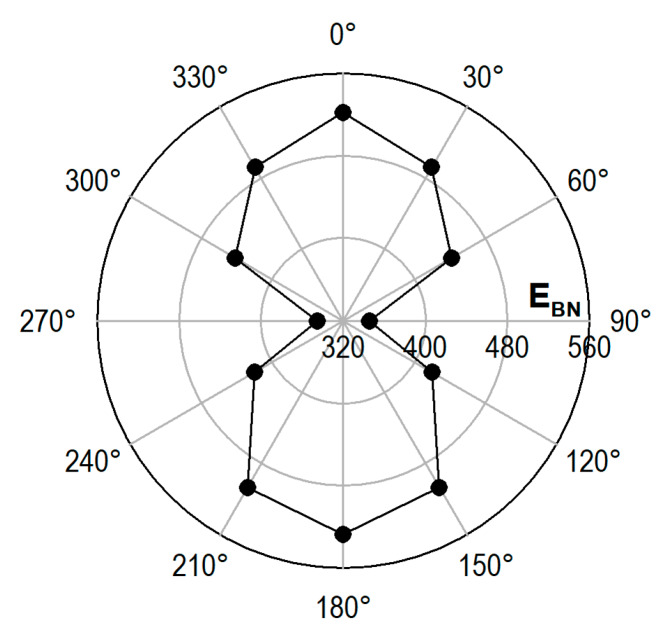
Polar diagram illustrating average Barkhausen noise energy *E_BN_* in the place of the specimen narrowing (0–180°—direction parallel to the tensile load direction, 90–270°—direction normal to the tensile load direction).

**Figure 7 materials-13-02059-f007:**
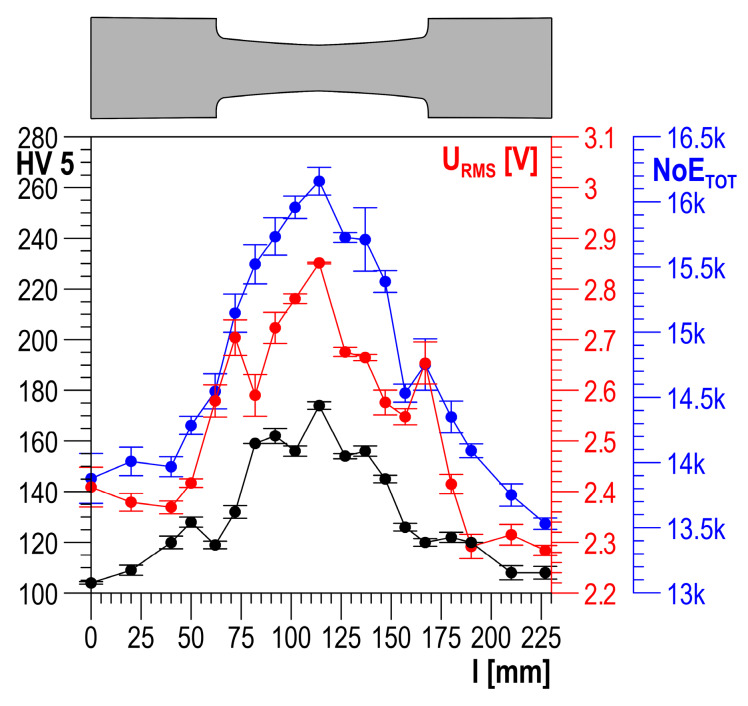
Distribution of HV5, root mean square (RMS), and *NoE_TOT_* along the S235 steel specimen length.

**Figure 8 materials-13-02059-f008:**
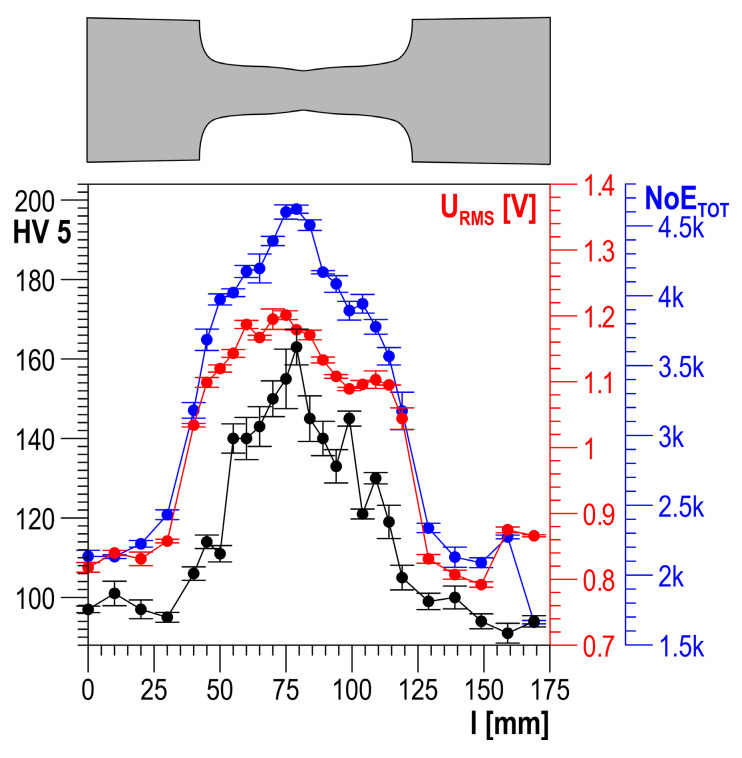
Distribution of HV5, RMS, and *NoE_TOT_* along the DC01 steel specimen length.

**Figure 9 materials-13-02059-f009:**
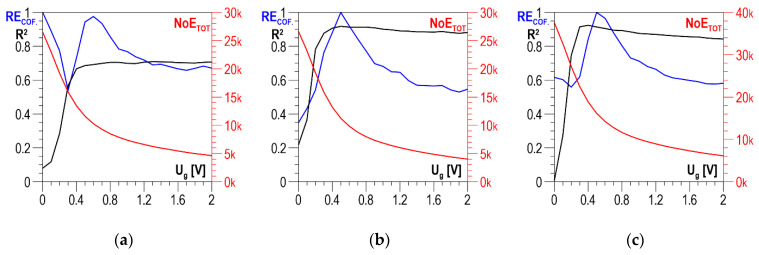
Averaged *NoE_TOT_* distributions and distributions of coefficients *RE_COF_* and *R*^2^ as a function of threshold voltage *U_g_*—S235 steel (**a**) parallel direction; (**b**) normal direction; (**c**) absolute value.

**Figure 10 materials-13-02059-f010:**
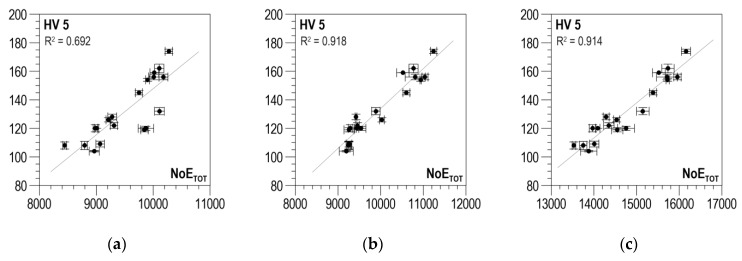
Linear correlations between *NoE_TOT_* and *HV5* hardness—S235 steel (**a**) parallel direction, *U_g_* = 0.6 V; (**b**) normal direction, *U_g_* = 0.5 V; (**c**) absolute value, *U_g_* = 0.5 V.

**Figure 11 materials-13-02059-f011:**
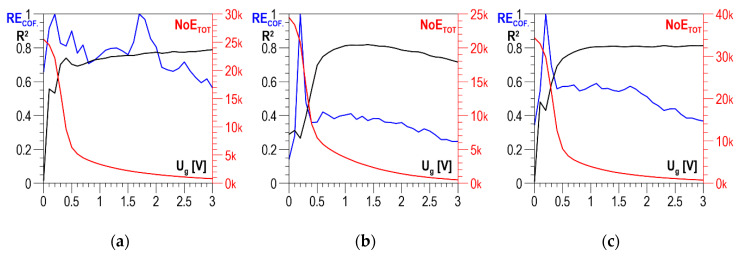
Averaged *NoE_TOT_* distributions and distributions of coefficients *RE_COF._* and *R*^2^ as a function of threshold voltage *U_g_*—DC01 steel (**a**) parallel direction; (**b**) normal direction; (**c**) absolute value

**Figure 12 materials-13-02059-f012:**
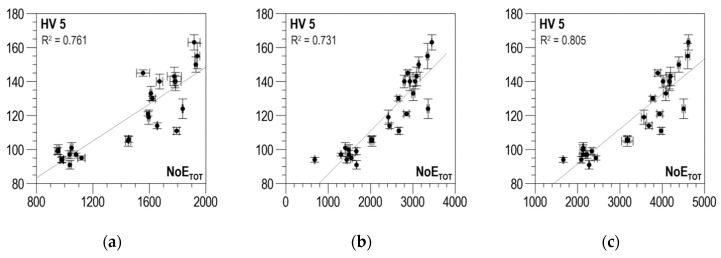
Linear correlations between *NoE_TOT_* and HV5 hardness—DC01 steel (**a**) parallel direction, *U_g_* = 1.7 V; (**b**) normal direction, *U_g_* = 1.1 V; (**c**) absolute value, *U_g_* = 1.1 V.

**Figure 13 materials-13-02059-f013:**
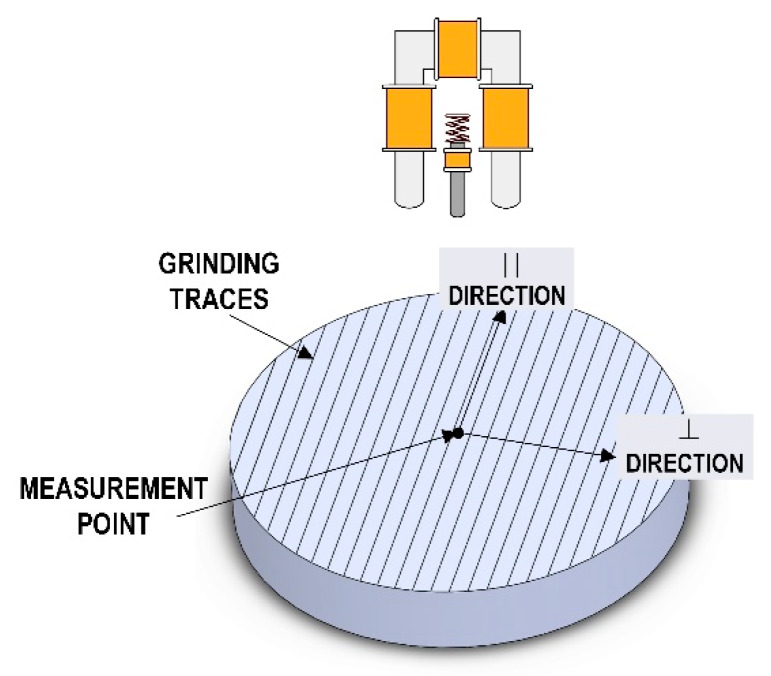
Details of the Barkhausen noise testing performed on specimens made of AMS 6414 steel.

**Figure 14 materials-13-02059-f014:**
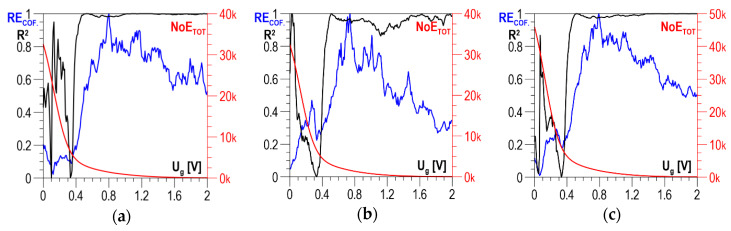
Averaged *NoE_TOT_* distributions and distributions of coefficients *RE_COF._* and *R*^2^ as a function of threshold voltage *U_g_*—Configuration C1 (**a**) parallel direction; (**b**) normal direction; (**c**) absolute value.

**Figure 15 materials-13-02059-f015:**
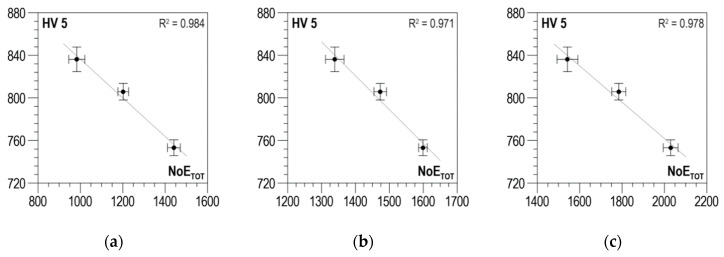
Linear correlations between *NoE_TOT_* and *HV5* hardness—Configuration C1 (**a**) parallel direction, *U_g_* = 0.8 V; (**b**) normal direction, *U_g_* = 0.74 V; (**c**) absolute value, *U_g_* = 0.79 V.

**Figure 16 materials-13-02059-f016:**
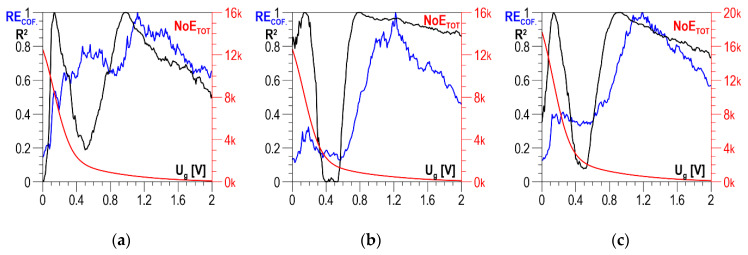
Averaged *NoE_TOT_* distributions and distributions of coefficients *RE_COF_* and *R*^2^ as a function of threshold voltage *U_g_*—Configuration C2 (**a**) parallel direction; (**b**) normal direction; (**c**) absolute value.

**Figure 17 materials-13-02059-f017:**
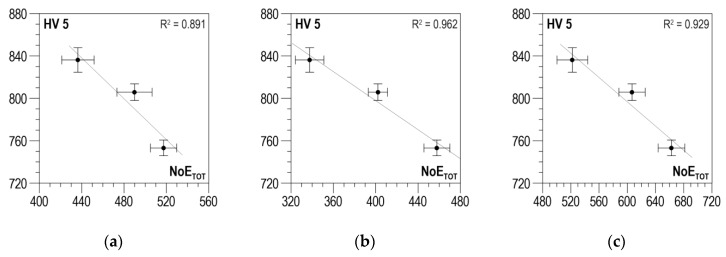
Linear correlations between *NoE_TOT_* and *HV5* hardness—Configuration C2 (**a**) parallel direction, *U_g_* = 1.12 V; (**b**) normal direction, *U_g_* = 1.22 V; (**c**) absolute value, *U_g_* = 1.19 V.

**Figure 18 materials-13-02059-f018:**
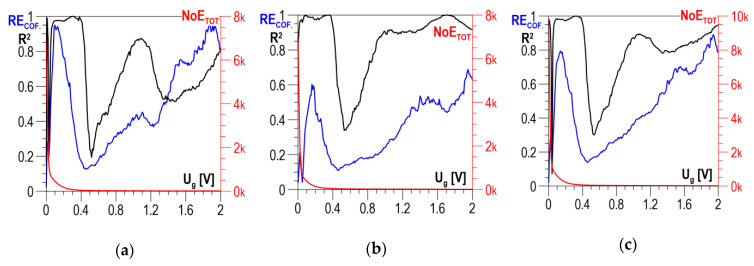
*NoE_TOT_* distributions and distributions of coefficients *RE_COF_* and *R*^2^ as a function of threshold voltage *U_g_*—Configuration C3 (**a**) parallel direction; (**b**) normal direction; (**c**) absolute value.

**Figure 19 materials-13-02059-f019:**
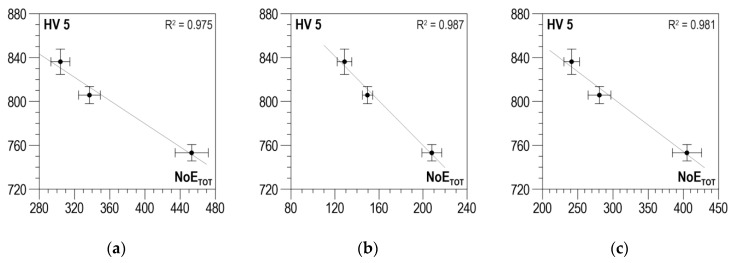
Linear correlations between *NoE_TOT_* and *HV5* hardness—Configuration C3 (**a**) parallel direction, *U_g_* = 0.1 V; (**b**) normal direction, *U_g_* = 0.16 V; (**c**) absolute value, *U_g_* = 0.14 V.

**Table 1 materials-13-02059-t001:** Steel chemical composition and strength properties.

	Chemical Composition Mass Fraction [%]						Strength Properties	
	C	Mn	P	S	N	Cu	Re [MPa]	Rm [MPa]
S235JR	0.21	1.4	0.035	0.035	0.012	0.55	235	410
DC01	≤0.12	≤0.60	≤0.045	≤0.045	-	-	140–280	270–410

**Table 2 materials-13-02059-t002:** Details of measuring configurations.

Configuration	S235 Steel	DC01 Steel
sampling frequency f_p_ [kHz]	800	800
magnetizing current amplitude [mA]	150	200
magnetizing current frequency [Hz]	2.02	2.02
signal amplification [dB]	21	35
number of cycles of changes in magnetization	5	5

**Table 3 materials-13-02059-t003:** Details of the measuring configuration.

Configuration	Configuration C1	Configuration C2	Configuration C3
sampling frequency f_p_ [kHz]	800	800	800
magnetizing current amplitude [mA]	200	200	200
magnetizing current frequency [Hz]	2.02	5.31	10.4
signal amplification [dB]	50	50	25
number of cycles of changes in magnetization	5	15	25

**Table 4 materials-13-02059-t004:** *HV 0.5* hardness measurement results.

Specimen	HV 5 Measurement Results										HV 0.5 Average Value	Standard Deviation
1	757	749	746	768	748	757	740	757	757	753	753.2	7.40
2	815	800	809	789	801	809	803	814	815	803	805.8	7.82
3	834	827	827	821	851	846	821	853	846	836	836.2	11.53

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
