# Peer review of "Evaluation of Ferromagnetic Steel Hardness Based on an Analysis of the Barkhausen Noise Number of Events"

_materials, 2020, doi:10.3390/ma13092059_

Round 1
Reviewer 1 Report
This study is valuable and I have found that could be published after minor revision. Authors investigate sample surface by the use of Barkhausen noise – number of detected pulses. The study also deals with evaluation of suitable threshold for MBN number counting. The idea of the paper is not new and pioneer work in this field was carried out about 10 years earlier (item 24 of authors list of references). However, authors introduced the modified approach. I recommend paper for publication after minor revision.
The main aspect which I found unclear or debatable are as follows:
- Authors indicate that “The specimens were repeatedly subjected to uniaxial tension in such a manner that with every loading cycle there was a slight increment in plastic strain. The loading process was discontinued the moment that a clear local narrowing appeared.” What kind of device do you used for plastic deformation of samples? Is it possible to obtain stress-strain curve for both materials? The abovementioned statement indicate that local narrowing appears. How the local narrowing (necking) was detected – by visual inspection. This is very debatable from the point of view of repeatability of experiments. The clear conditions of plastic deformation should be stated in the form of plastic strain.
- The samples in the first part of experiments are investigated in the area of non-homogenous plastic deformation – beyond stagnation point in stress-strain curve. This information also proves profile of hardness within the samples length. Please make it clear.
- Metallography images of the tested samples would be beneficial – optional comments.
- V – parameter in page 7 – is it rms value or energy of MBN?
- DWs motion occur in the form of avalanches – the number of the pulses therefore depend on sampling frequency of MBN signal as well as the ability of magnetic field to unpin DWs. Please mention that this in your manuscript – especially sampling frequency can take remarkable role since some pulses could overlap in the time scale.
- Please provide physical background associated with increasing number of pulses versus plastic deformation, heat treatment and DW alignment, density, etc.
- Please provide rms values of MBN for all measurements since rms is a function of MBN number of pulses and their magnitude. I would expect that rms value and magnitude of pulses would decrease along with increasing plastic deformation and hardness together with increasing number of pulses. Try to investigate this aspect.
- Considering the second part of your experiments grinding process decrease the true case-hardened region as compared with heat treatment. Moreover, your way of measurement of hardness is not correct. Case-carburized samples hardness is investigated via microhardness technique since progressive decrease of hardness occurs from the surface towards the bulk. Microhardness is the usually measured in the cross sectional views – this point of my review is only comment for the future research. You can leave your data as they are in the revised version.
- List of references contains mistakes for instance item 21
There were published other reports in which number of MBN pulses were employed for monitoring of surface such as:
Reviewer 2 Report
List of minor comments:
Line 32: replace the term "fragment of the surface" by "surface domain".
Line 60: acronym "RMS" should be explicitely indicated at this line.
Line 63: delete "root-mean-square".
Line 93: "Berkhausen noise", the authors seem to consider Berkhausen noise and effect as same physical entities. If this is the case, please make sure to use the same terminology through out the manuscript.
Line 95: replace "rms" by "RMS", make sure to use the same capital letter acronym through out the manuscript.
Line 101: "rms", see comment Line 60.
Line 117: "obtained very good results", replace the expression "very good" by "encouraging" or "promising" results.
Line 145: "noise detection, magnetization control", replace the comma by "and".
Line 159: "processing process produces", replace "process" by "method" or "procedure".
Line 176 and 178: the acronym of the "discrimination voltage" should be written using the same police.
Line 193: "by 4÷5": the significance of the fraction symbol is not clear. Do the authors mean the ratio "4/5". Please change your notation.
Line 201 and Table 1: the steel reference should be the same.
Line 207: "a clear local narrowing appeared": How did the authors evaluate this clear narrowing? based on visual inspection? please elaborate.
Equation 3: the notations of the indices "x" and "Y" should be in the came police.
Line 241: remove the word "below" and indicate instead the inmage number.
Line 320: The sentence is not complete, it seems that the sitation of Table 3 in the text is also missing at this level.
Line 322: caption of Figure 13, add a space "onspecimens".
Lines 335 and 336: the authors should present the figures without using tables of figures captions. This is does not seem to be adequate in sceintific papers.
Line 461: ref. 21, check the authors names "Mart??nez-De-Gueren";
Line 487: The reference title should be in a sentence form, with a capital letter limited to the first letter only
Line 489: missing bracket.
Line 492: same comment as in Line 487.
Reviewer 3 Report
This manuscript proposed a new method to evaluate the hardness of ferromagnetic steel based on analysis of barkhausen effect number. Overall, this is a very interesting manuscript, which was well organized and written. I suggest that it can be published in Materials if the authors can well address the following comments. 1. Please illustrate the main innovation of this research. Why is Barkhausen effect total number of events considered for the task of interests? What is superiority over other methods? 2. There are several other methods to evaluate the hardness. It will be better if the authors can compare the proposed method with 1-2 other methods to evaluate the performance. 3. More future research should be included in conclusion part.Author Response
Please see the attachment.
